# Brief communication: Updated grounding line mapping in the Amundsen Sea Embayment, Antarctica, from 1-day repeat Sentinel-1 SAR data

Jonas K. Andersen<sup>1</sup>, Romain Millan<sup>2</sup>, Eric Rignot<sup>3,4</sup>, Bernd Scheuchl<sup>3</sup>, Jean Baptiste Barré<sup>3</sup>, and Anders A. Bjørk<sup>1</sup>

**Abstract.** Knowledge of Antarctic glacier grounding lines, which mark the transition between grounded and floating ice, is a vital parameter in determining the stability of major ice shelves and hence the ice sheet. Rapid grounding line retreat and associated mass loss has been documented at numerous Antarctic glaciers, particularly in the Amundsen Sea Embayment. However, few comprehensive grounding line mappings exist, particularly from recent years. Here, we utilize a unique record of Sentinel-1 Synthetic Aperture Radar 1-day repeat-pass imagery to generate a comprehensive retrieval of grounding line location in the Amundsen Sea Embayment in 2025 and evaluate recent changes.

## 1 Introduction

Mass loss from the Antarctic Ice Sheet has increased over the past four decades, mainly due to speed-up and increased ice discharge of glaciers in West Antarctica (Otosaka et al., 2023). Glacier acceleration is linked to ice shelf weakening, driven by the intrusion of warm, saline circumpolar deep water beneath floating ice, which enhances basal melting (Holland et al., 2023). Melting peaks near the grounding line (GL), the boundary between grounded and floating ice. As temperatures rise, GL retreat into deeper basins accelerates ice flow and dynamic thinning, further promoting retreat and decreasing the buttressing potential of ice shelves (Schmidtko et al., 2014; Holland et al., 2023; Joughin et al., 2014). In the Amundsen Sea Embayment (ASE), located at the West Antarctic Ice Sheet (Figure 1) and holding a 1.2 m potential sea level rise, monitoring of grounding lines is of particular interest due to their documented rapid retreat into drainage basins deep below sea level, which will likely lead to further instabilities and mass loss in the future (Mouginot et al., 2014; Park et al., 2013; Rignot et al., 2014; Scheuchl et al., 2016).

Given the grounding line's critical role in Antarctic ice sheet stability and mass loss, accurate information on its evolution is essential for better constraining ice-ocean interactions and predicting the ice sheet's future evolution and its contribution to sea-level rise. Thus, extensive and repeated observations of this critical boundary are essential (Konrad et al., 2018; Rignot, 2023). The need for frequent observations is compounded by the fact that the true GL fluctuates during a tidal cycle within

<sup>&</sup>lt;sup>1</sup>Department of Geosciences and Natural Resource Management, University of Copenhagen, 1350 Copenhagen, Denmark

<sup>&</sup>lt;sup>2</sup>Univ. Grenoble Alpes, CNRS, IRD, INRAE, Grenoble-INP, IGE (UMR 5001), 38000 Grenoble, France

<sup>&</sup>lt;sup>3</sup>Department of Earth System Science, University of California, Irvine, CA 92697

<sup>&</sup>lt;sup>4</sup>Radar Science and Engineering Section, Jet Propulsion Laboratory, California Institute of Technology, Pasadena, CA 91109 **Correspondence:** Jonas K. Andersen (joka@ign.ku.dk)

a grounding zone, which may be several kilometers wide, depending on factors such as tide magnitude, ice thickness, and bedrock slope (Fricker and Padman, 2006).

While in-situ methods for observing grounding line location have been demonstrated (e.g., Le Meur et al. (2014)), remote sensing techniques offer a more feasible alternative for large-scale, repeated retrievals. Several remote sensing techniques have been applied including phase-based and amplitude-based processing of satellite Synthetic Aperture Radar (SAR) data, repeat laser or radar altimetry, and optical imagery - a comprehensive review of these methods are provided by Friedl et al. (2020). The most accurate retrievals are obtained from double-difference SAR interferometry (Rignot, 1996; Friedl et al., 2020) (described in section 2.2). This technique has been used to map grounding lines across the Antarctic Ice Sheet, with the ERS-1/2 satellites allowing for a particularly extensive coverage, however only during periods where the satellites flew in short repeat-pass constellations (i.e., the 1-day tandem phases during 1995-1996 and 1999-2000 and the 3-day ice phases during 1991-1992, 1994, and 2011) (Rignot et al., 2016). While the temporal resolution of these acquisitions was limited to discrete intervals, precluding a full delineation of the grounding zone, the ERS imagery enabled the detection of rapid and widespread grounding line retreat across West Antarctica during the 1991-2011 period (Rignot et al., 2014). Recent advances using data from the Sentinel-1 archive provided an estimate of the grounding zone, through a dense time series of automated interferometry-based GL retrievals from 2018 (Mohajerani et al., 2021). However, this retrieval was limited by the relatively long repeat-pass period of Sentinel-1 (6 or 12 days), hindering delineations in the fastest-changing sectors, where strong decorrelation occurs in central glacier trunks experiencing the highest GL retreat. Some previous studies have acquired commercial/non-public SAR data with short repeat-pass periods to provide well-resolved GL retrievals over specific glaciers (e.g., Rignot et al. (2024); Milillo et al. (2022)). However, no public, routinely acquired SAR data with a repeat-pass period short enough to provide adequate GL delineations over the fastest-changing glaciers in the ASE currently exists.

Here, we use Sentinel-1 imagery with a 1-day repeat-pass period, acquired during January-March 2025 for the in-orbit commissioning of Sentinel-1C, to delineate glacier grounding lines across the Amundsen Sea Embayment. The short repeat-pass data allows for a well-resolved, contemporary delineation of ASE grounding lines, most of which have been mapped only rarely in the past decade and, to our knowledge, not at all since 2020 or earlier.

# 2 Data and methods

45

## 2.1 Sentinel-1 data

The EU Copernicus Sentinel-1 satellite constellation nominally consists of two C-band SAR satellites, orbiting in a polar, sunsynchronous orbit 180 degrees out of phase. The orbit repeat-pass period is 12 days, yielding a 6-day repeat-pass period with two active satellites. Sentinel-1A, launched in April 2014, remains active, while Sentinel-1B (launched in July 2016) ceased operations on 23 December 2021 due to a power system failure. Consequently, dense 6-day repeat-pass coverage was available across the Antarctic and Greenland ice sheet margins from 2016 to 2021. From 2022 to early 2025, the constellation operated with only Sentinel-1A, yielding 12-day temporal baselines. The launch of Sentinel-1C on 5 December 2024, followed by its

in-orbit commissioning (completed in May 2025), restored 6-day repeat coverage. Sentinel-1D is currently planned for launch in late 2025 to replace 1A, which has exceeded its expected lifetime.

A short temporal baseline (i.e., the time separation of images) in interferometric SAR processing is vital, as increased baselines generally lead to increased decorrelation. In many parts of the marginal Antarctic and Greenland ice sheets, high flow speeds, snowfall and redistribution, and/or surface melt yield total decorrelation for 6- or 12-day Sentinel-1 image pairs all year, excluding the retrieval of flow speeds or grounding line delineations.

During the in-orbit commissioning phase of Sentinel-1C, the satellite was temporarily placed in a 1-day offset orbit relative to Sentinel-1A from January 17th to March 7th 2025. A total of 166 Single Look Complex image slices, covering four orbit tracks (see Figure S1) in the Interferometric Wide swath mode, were acquired from both satellites over the marginal Antarctic Ice Sheet in the Amundsen Sea Embayment (Figure 1). These 1-day repeat retrievals form the basis of our updated 2025 grounding line delineation.

# 65 2.2 Double-difference interferometry for grounding line detection

Differential SAR interferometry (DInSAR) measures the phase difference between two subsequent SAR acquisitions, which, after correcting for satellite geometry and surface topography (using a Digital Elevation Model), is proportional to surface displacement in the radar line-of-sight (LoS) direction (Massonnet et al., 1993). Because the LoS vector has both vertical and horizontal components, phase changes can reflect motion in either direction:

$$70 \quad \Delta\phi_{LoS}^1 = \Delta\phi_{horz}^1 + \Delta\phi_{vert}^1 \tag{1}$$

By differencing two sequential DInSAR measurements, a technique known as double-difference interferometry, we isolate changes in LoS displacement between the two time intervals. If horizontal velocity remains steady, these differences primarily reflect changes in vertical displacement:

$$\Delta\phi_{LoS}^{DD} = \left(\Delta\phi_{horz}^2 + \Delta\phi_{vert}^2\right) - \left(\Delta\phi_{horz}^1 + \Delta\phi_{vert}^1\right) = \Delta\phi_{vert}^2 - \Delta\phi_{vert}^1$$
 (2)

Over floating ice shelves, tidal variations between acquisitions often produce different vertical motion contributions between repeat passes, leading to measurable double-difference phase signals in the form of dense fringes, while the (presumed) constant horizontal flow contribution cancels out. The inland limit of these fringes marks the limit of tidal flexure of the ice, which approximately coincides with the grounding line, although the true grounding line will generally lie slightly seaward of the flexure limit (Fricker and Padman, 2006). This approach is widely regarded as one of the most accurate remote sensing methods for grounding line detection and has been applied with various SAR sensors (Rignot, 1996; Joughin et al., 2010; Friedl et al., 2020).

Differential interferograms were processed using the workflow outlined in Andersen et al. (2020). The REMA DEM at 100 m resolution (Howat et al., 2022) and MEaSUREs Antartica Ice Velocity product at 450 m resolution (Rignot et al., 2017)

were applied in the refined coregistration procedure, and all images from each respective track were resampled to the same reference image. We use Precise Orbit Ephemerides (POE) to update orbit state vectors only for Sentinel-1A acquisitions, as POE products for Sentinel-1C were not available for this period. Finally, double-difference interferograms are then formed simply by differencing the phase images of sequential interferograms.

Grounding lines were manually digitized at the inland limit of the tide-induced fringe patterns (see Figure 2). For some areas, multiple double-difference interferograms are available due to multiple acquisition cycles from the same track or overlapping of different orbit tracks. In those cases, we generally select the most coherent retrieval, that best resolves grounding line features, for delineating the grounding line. We also digitize pinning points, i.e. localized areas of the ice shelf that stick to bathymetric highs and act to buttress and stabilize ice flow.

## 3 Results

Figure 1 shows an overview of the 2025 Amundsen Sea Embayment grounding line from the Sentinel-1 1-day repeat imagery. The delineation spans from the Abbot Ice Shelf to the Getz Ice Shelf. Figure 2 shows Sentinel-1 1-day double-difference interferograms for the four regions highlighted in Figure 1. The interferograms are generally highly coherent and a contiguous grounding line is delineated for nearly the entire coast, with a few exceptions in zones of high velocity gradients at Thwaites, Pine Island, and Kohler glaciers, which lead to decorrelation even with the 1-day repeat pass period.

Figure 3 shows a comparison between the new 2025 grounding line delineation and the MEaSUREs Antarctica grounding line product (Rignot et al., 2016), which contains retrievals from the period 1992-2014, the ESA CCI grounding line product (ESA AIS CCI, 2021), containing retrievals from the period 1994-2020, and the COSMO-SkyMed grounding line dataset from Milillo et al. (2022), covering the Pope, Smith, and Kohler Glaciers during 2016-2020, depending on which product contains the latest well-resolved delineation for the given glaciers. The grounding lines are overlaid on the BedMachine v3 bed elevation product (Morlighem, 2022). In the ASE sector, estimates of grounding zone width are mostly around 1 km but may locally exceed 5 km (Mohajerani et al., 2021). Consequently, the reported changes in GL location should be viewed as approximative, as they stem from a comparison of discrete acquisitions.

At Abbot Ice Shelf, which extends from the Bellingshausen Sea into the ASE, the GL remains almost completely unchanged since the 1990s, with the 2025 delineation lying within  $\pm 1$  km of the 1992/1995 position almost everywhere (Figure 3a). Christie et al. (2016) found a widespread but modest grounding line retreat during 1990-2015 (

flowing trunk, where the 2025 GL lies 1-2 km inland from the 2011 delineation (Figure S2a-b). The apparent stability is not surprising, as the Cosgrove grounding line is situated on a predominantly prograde bed (Figure S2b).

The fastest-flowing sector of the Getz Ice Shelf is not covered by the Sentinel-1 1-day repeat data set. For the rest of the ice shelf, we compare the 2025 GL delineation with the 2018 retrievals from Mohajerani et al. (2021) and note that the 2025 GL lies approximately within the estimated 2018 grounding zone, with a few local exceptions, in which the 2025 GL lies 1-2 km inland (Figure S2c-d).

At Pine Island Ice Shelf, the main trunk grounding line has shown a further retreat of approximately 2-7 km since 2011, following a larger retreat of 15-20 km during 1992-2011. In the main trunk of the glacier - the region with the fastest ice flow - the western section has retreated by up to 7 km, while the central part has pulled back by approximately 2.5 km. The western section of the main trunk appears to have been dislodged from a sill in the bedrock topography at a depth of -1000 m and has retreated by around 5 km (Figure 3b). In this critical part of the glacier, where ice discharge is at its maximum, the GL has retreated more significantly along the eastern and western flanks than at the center. The eastern branch of Pine Island Glacier, which flows toward the ice shelf front, has also experienced a grounding line retreat averaging around 5 km, with localized retreats reaching up to 7 km, particularly in areas where the bedrock slope is slightly retrograde. On the western flank of the ice shelf, however, the grounding line has remained relatively stable, situated on a more pronounced, mountainous, and prograde topography (Figure 3b).

The Thwaites Ice Shelf grounding line has shown a spatially varying retreat since 1992, ranging from 

The Kohler Glacier grounding line retreated 8 km during 1992-2011, re-advanced 4.5 km during 2011-2014, then retreated 7 km during 2014-2020. The 2025 grounding line now sits approximately at the 2018-2020 position, about 3 km from the top of a steep prograde slope (Milillo et al., 2022).

The abated grounding line retreat during 2018-2025 of Pope, Smith, and Kohler (compared to the preceding decades) is associated with a shift in the ice flow regime: Selley et al. (2025) observed near-steady flow speeds during the 2015-2022 period for all glaciers, whereas all glaciers showed rapid flow acceleration for all or parts of the 2005-2015 period.

Additionally, The Dotson ice shelf also appears to have lost two pinning points since 2016. Furthermore, many smaller glaciers in the region, including Philbin Inlet, Singer Glacier, and McClinton glaciers, also show grounding line retreats in the range of 2-6 km since their latest delineation from the MEaSUREs dataset, although their 2025 GL positions do appear to lie within (or near) the estimated 2018 grounding zones from Mohajerani et al. (2021).

#### 4 Conclusions

Continually updated retrievals of glacier grounding lines are essential for understanding the ongoing rapid mass loss from the ASE as well as detecting early signs of retreat and instability at other ice shelves (Li et al., 2023; Brancato et al., 2020; Milillo et al., 2019; Millan et al., 2022). Existing spatially comprehensive GL products are based primarily on data from ERS-1/2, Sentinel-1, and TerraSAR-X/TanDEM-X with additional retrievals from Radarsat-1/2, ALOS PALSAR, and COSMO-SkyMed (Rignot et al., 2016; Mohajerani et al., 2021; Milillo et al., 2022; ESA AIS CCI, 2021). The vast majority of well-resolved, contiguous grounding line delineations, however, come from short repeat-pass SAR acquisitions, such as ERS-1/2 imagery from the tandem mission phase (1-day repeat-pass, 1995-1996 and 1999-2000) and the ice phase (3-day repeat-pass, winters of 1991/1992 and 1993/1994 and 2011) (Rignot et al., 2016; Friedl et al., 2020). Other studies have used short repeat-pass data from non-public/commercial SAR satellites, allowing for highly resolved GL retrievals at specific glaciers (e.g., Milillo et al. (2019); Rignot et al. (2024)). No current satellite mission provides publicly available short (

**Figure 1.** Overview of Sentinel-1 2025 Amundsen Sea Embayment grounding line delineation (black line), overlaid on an ice velocity mosaic from MEaSUREs (Rignot et al., 2017). Black rectangles indicate spatial extents of panels (a)-(d) in Figures 2 and 3.

**Figure 2.** Sentinel-1 1-day repeat double-difference interferograms from 2025 covering Abbot Ice Shelf (a), Pine Island Ice Shelf (b), Thwaites and Haynes glaciers (c), Pope, Smith, and Kohler glaciers (d). Acquisition dates and orbit track numbers are provided in Table S1 in the Supplementary Material. The 2025 Sentinel-1 grounding line product is indicated by black lines. The dense fringe patterns arise from differences in the tide-induced vertical flexure between grounded and floating ice. The inland limit of this flexure is interpreted as the (approximate) grounding line. The interferograms also reveal pinning points (in the form of local, circular fringe patterns).

**Figure 3.** Grounding line changes at Abbot Ice Shelf (a), Pine Island Ice Shelf (b), Thwaites and Haynes glaciers (c), Pope, Smith, and Kohler glaciers (d). Background map shows bed elevation from BedMachine v3 (Morlighem, 2022). The 2025 Sentinel-1 grounding line product is indicated by orange lines, while the MEaSUREs (Rignot et al., 2016), ESA CCI (ESA AIS CCI, 2021), and Milillo et al. (2022) grounding line products are indicated by various colors, depending on the retrieval year (see legend in panel (a)).

ongoing retreat at glaciers such as Pine Island and Thwaites as well as the potential onset of grounding line retreat, and hence instability, at other ice shelves such as Abbot, Cosgrove, and Getz.

The 2025 grounding line retrieval highlights areas of continued vulnerability, particularly at Pine Island and Smith Glaciers, where the grounding lines are situated at or near retrograde bed slopes. At Pine Island, retreat into deeper terrain will further enhance discharge, while continued retreat at Smith could soon lead to renewed instability as the glacier enters another retrograde section of the bed. These configurations emphasize the value of high-resolution, repeated grounding line mapping to track evolving glacier stability.

Data availability. The 2025 Amundsen Sea Embayment grounding line product, along with geocoded Sentinel-1 double-difference interferograms used for delineations, is available at https://doi.org/10.5281/zenodo.17019510. Sentinel-1 imagery, including Precise Orbit Ephimeredes, is available at https://dataspace.copernicus.eu/ (Rignot et al., 2016). The MEaSUREs Antarctica grounding line product is available at https://doi.org/10.5067/IKBWW4RYHF1Q and the ESA CCI Antarctica grounding line product is available at https://climate.esa.int/en/projects/ice-sheets-antarctic/ (ESA AIS CCI, 2021). The MEaSUREs Antarctica velocity mosaic is available at https://doi.org/10.5067/D7GK8F5J8M8R (Rignot et al., 2017), the BedMachine v3 bed elevation product is available at https://nsidc.org/data/nsidc-0756/versions/3 (Morlighem, 2022), and the REMA Digital Elevation Model is available at https://doi.org/10.7910/DVN/EBW8UC (Howat et al., 2022).

*Author contributions.* J.K.A., A.A.B., and E.R. designed the study. J.K.A. and J.B.B. carried out data processing, with analysis contributions from all authors. J.K.A wrote the initial draft of the manuscript, with editing from all other authors.

00 Competing interests. The authors declare no competing interests.

Acknowledgements. We thank Nuno Miranda and ESA for publicly providing the 1-day repeat Sentinel-1 data. J.K.A. and A.A.B. acknowledge support from the Villum Foundation (Villum Young Investigator grant no. 29456) and the Independent Research Fund Denmark Sapere Aude Research Leader grant (10.46540/2064-00050B). R.M acknowledges support from the European Research Council (ERC) under the Horizon Framework research and innovation program of the European Union (grant no. 101164392; "IceDaM" project). The work was partially performed at the University of California Irvine and Caltech Jet Propulsion Laboratory under a contract with the National Aeronautics and Space Administration Cryosphere Science (80NSSC23K0177), MEaSUREs (80NSSC23M0146), and MAP (80NSSC20K1076) programs and the National Science Foundation Thwaites-MELT (1739003).

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
