# Peer review of "Brief communication: Updated grounding line mapping in the Amundsen Sea Embayment, Antarctica, from 1-day repeat Sentinel-1 SAR data"

_EGUsphere, 2025_

## Referee Comment (RC2)

*Review of:*

***Andersen et al. (2025) Brief communication: Updated grounding line mapping in the Amundsen Sea Embayment, Antarctica, from 1-day repeat Sentinel-1 SAR data***

**Summary:**

In this brief communication, the authors present a new, near-continuous grounding line (GL) delineation for the Amundsen Sea Embayment (ASE) derived from a unique set of 1-day repeat-pass Sentinel-1 SAR data acquired during the 2025 Sentinel-1C in-orbit commissioning phase. The short repeat interval represents a rare opportunity to overcome the severe coherence limitations that typically hinder InSAR-based GL mapping in this region, caused by fast flow, heavy crevassing, surface melt, snowfall, and rapid GL retreat. As a result, the study provides an important and timely update to existing GL products in the region, which in places have not been observed for several years. The authors find minimal GL change at Abbot, Cosgrove, and Getz ice shelves since they were last observed, while Pine Island Glacier has retreated by approximately 2–7 km since 2011. At Thwaites Glacier, the 2025 grounding line is largely located within the grounding zone region identified in recent ICEYE observations (Rignot et al., 2024) and also shows evidence of upstream seawater intrusions.

**Overall assessment:**

This is a well-written and timely manuscript that presents valuable new insights alongside a high-quality dataset of clear relevance to the wider glaciological community. In particular, this study demonstrates the substantial improvement in GL detection achievable with 1-day repeat-pass synthetic aperture radar data and provides an important, up-to-date GL dataset for the Amundsen Sea Embayment; one of the most rapidly evolving sectors of the Antarctic Ice Sheet. I recommend this manuscript for publication, subject to the authors addressing one general (moderate) concern regarding uncertainties associated with tidal GL migration, along with several minor suggested edits to the text and figures to improve clarity, as detailed below.

**General comments:**

The interpretation of the new GL dataset would benefit from a more explicit discussion of how differences in coincident sea surface height (SSH) at the time of each SAR acquisition – driven by tides and dynamic atmospheric contributions – affect the vertical deflection and GL position recorded in each double-difference interferogram. This would help clarify how tidal GL migration may influence the reported long-term GL retreat rates and better constrain the associated uncertainties, which I appreciate are difficult to explicitly quantify. Several of the reported GL retreat distances are comparable to grounding zone (GZ) widths (of order 1–5 km) inferred in previous studies, making the tidal context of each new GL delineation particularly important for interpretation.

Specifically, I recommend that the authors provide the precise acquisition time (not just the date) and corresponding SSH for each SAR image, as well as the maximum SSH and relative SSH difference between the four images used to form each interferogram. Previous studies have emphasized the importance of presenting new GL products alongside this contextual metadata (e.g. Freer et al., 2023), and this could be readily addressed by adding the information to Table S1. Additionally, it would be useful to comment on whether – where multiple interferograms are available from repeated acquisition cycles along the same track – there are any signals of tidal GL migration that could help to constrain variability in GZ width.

**Specific comments:**

- **Abbreviations -** Please ensure consistent use of the abbreviated term "GL" throughout the manuscript. The text currently alternates between "grounding line" and "GL". The same applies to any other abbreviations.
- **Capitalisation -** Please ensure consistent capitalization of "glacier(s)" and "ice shelf/ice shelves", particularly where plurals are used. For example, "Pope, Smith, and Kohler Glaciers" (L102) versus "Pope, Smith, and Kohler glaciers" (L98, L142), and "Crosson and Dotson Ice Shelves" (L142) versus "Dotson ice shelf" (L157). While this may ultimately be governed by journal style, I believe that the general convention is to capitalize singular proper names (e.g. "Pine Island Glacier", "Dotson Ice Shelf") and to use lower case for plural or generic references.
- **Grounding zone definition -** Please clarify how the grounding zone (and its width) are defined. In the existing literature, the term has been used to describe both the full region experiencing tidal flexure (i.e. between Points F and H) and the distance over which the grounding line migrates with the tide (sometimes termed the "ice grounding zone") – the authors seem to use the latter definition, but it would be useful to state explicitly.
- **L116 –** Add a reference to Fig S2a-b in this first sentence where the results for Cosgrove Ice Shelf are first described.
- **L123-132 –** The description of the retreat patterns across Pine Island Ice Shelf are difficult to follow, I think due to the discrepancy between true vs grid compass directions. This section would benefit from some text refinement and addition of some more descriptive labels in Fig. 3b.
  - **L125** – The text here states that the western section of the main trunk has retreated by up to 7km, followed by a statement that the western section of the main trunk […] has retreated by around 5 km. Is this latter sentence meant to refer to the eastern section instead, or perhaps to the adjacent glacier trunk (grid west) on the other side of the bedrock high?  Adding labels to Fig 3b would help to clarify this here.

- ○ **L127** – Please clarify what is meant by 'in this critical part of the glacier'?
- ○ **L128** – It is unclear what is meant by the 'eastern branch of Pine Island Glacier'. If this refers to the tributary glacier (sometimes referred to as 'Piglet'), please state this explicitly and add labels to Fig. 3b.
- **L135** – The authors state that the 2025 GL lies within the 2023 GZ, but this is not shown in Fig 3c. Is there a reason why the 2023 ICEYE grounding zone has not been included on the map? As currently presented, Fig. 3c gives the impression of substantial GL retreat since the last available observations (ESA CCI 2016), which is potentially misleading without the 2023 ICEYE context.
- **L138** – These seawater intrusions are important observations and are clearly visible as bulls-eye patterns in the interferogram in Fig 2c. I suggest adding labels or dotted outlines to highlight these features and aid interpretation. While the comparison with Rignot et al. (2024) in Fig. S3 is informative, not all readers will consult the supplementary material, so additional annotation in Fig. 2c would strengthen the main text.
- **L140** – Do the regions that you have observed higher retreat rates coincide with these subglacial discharge outflows? If so, this is an interesting finding, and these features should be labelled on Fig. 3c. If not, this sentence feels somewhat out of place and disconnected from the otherwise descriptive results presented in this paragraph.
- **L157** – It is not clear from Fig 3d where the two pinning points reported as having been lost since 2016 on Dotson Ice Shelf are located. To me it appears that only the smaller pinning point closer to the main GL (grid east) is no longer grounded according to the updated 2025 GL. Please clarify this point and/or annotate the figure accordingly.
- **L158** – The GL retreat observed at these smaller glaciers is interesting, but their locations are missing from any of the figures. Please add their locations to Fig. 1a and consider adding supplementary figures to show these results.

**Figure comments**

- Please add North arrows to all figure panels with maps. This is particularly important given that true and grid north are often opposite in this region, which makes it difficult to interpret the results described in the main text.

**Figure 1**

- Please add additional labels with the names of the major glaciers/ice shelves highlighted in each box (i.e. Abbot, Pine Island, Thwaites, Dotson/Crosson), as well as the other glaciers referenced in the text (e.g. Philbin Inlet, Singer Glacier, McClinton glaciers).
- I also suggest adding inset boxes for Cosgrove and Getz ice shelves, which are discussed in the manuscript and shown in Fig S2.

**Figure 2**

- Consider changing the colour of the labels for 'Pine Island Ice Shelf', 'Thwaites Glacier' and 'Crosson Ice Shelf', as the current white text is difficult to read against the background.
- Fig 2d – Please remove the stray light blue '2016' label at Pope Glacier.

**Figure 3**

- Please adjust either the colour scale of the MEaSUREs/CCI grounding lines or the background bathymetry, as the current colours are very similar and make the grounding lines – particularly those shown in light green – difficult to distinguish.
- Please add a line showing the location of the ice shelf fronts where they fall within the mapped domain.
- **Fig 3b** – Please add additional descriptive labels such as 'main trunk', 'tributary', 'western branch' etc. to match the terminology used in the main text.
- **Fig 3c** – Please add the 2023 ICEYE GZ region to the map, and label both the seawater intrusions (those seen here and by Rignot et al., 2024) and any prominent ice shelf basal channels, to support the statements discussed in L134-140.
- **Fig 3d** – Please remove the stray light blue '2016' label at Pope Glacier.

**Figure S2 -** Please add a citation for the BedMachine v3 bed elevation dataset in the caption.

**Table S1 -** Please add the time stamp and coincident SSH information for each SAR image and interferogram, as requested above.